# Research on the Causes of Gas Explosion Accidents Based on Safety Information Transmission

**DOI:** 10.3390/ijerph191610126

**Published:** 2022-08-16

**Authors:** Huimin Guo, Shugang Li, Lianhua Cheng

**Affiliations:** College of Safety Science and Engineering, Xi′an University of Science and Technology, Xi’an 710054, China

**Keywords:** gas explosion, safety information transmission, safety information loss, accident-causation model, DEMATEL-ISM

## Abstract

To better explain the cause of gas explosion accidents, based on the existing accident-causation theory, this paper proposes an accident-causation model of gas explosion accidents based on safety information transmission. Based on this, a new method for the prevention of gas explosion accidents can be developed. By analysing the connection between safety information transmission and the causal factors of gas explosion accidents, it is inferred that the loss of safety information transmission is the key factor leading to accidents. Safety information transmission is a process chain in which information is transmitted between the information source and information subject. This process involves the stages of information generation, conversion, perception, cognition, decision-making, and execution. Information loss is inevitable during the transmission process. When the information loss of the degree of safety affects the judgment of the information subject on the current situation and decision making, the possibility of accidents increases. Therefore, in this study, we constructed an accident-causation model for gas explosion accidents based on the three elements and six stages of safety information transmission. Subsequently, the DEMATEL-ISM method was used to quantitatively analyse the causes of gas explosion accidents. Through a multilevel hierarchical structure division of the accident causes, the cause, result, and root factors affecting accidents were identified, and countermeasures were proposed to provide a theoretical basis for the prevention of gas explosion accidents.

## 1. Introduction

Various types of accidents occur in coal mines, including gas and coal dust explosions, coal and gas outbursts, suffocation accidents, fires, floods, roof falls, and haulage accidents. Gas explosions are undoubtedly the most serious type of accidents [1,2,3]. Once a gas explosion occurs, it can cause a large number of casualties, huge economic losses, and severe negative societal effects. According to yearly statistics from the Chinese National Coal Regulatory Administration, the vast majority of major accidents in Chinese coal mines are gas explosions. The risk of gas explosions is widespread in mines owing to the associated characteristics of gas and coal. At the same time, in the process of coal mining, the residual coal in the goaf, working face and exposed coal wall continuously desorb the gas and release it into the underground space, which determines the dynamic, random, and other complex characteristics of the gas accumulation phenomenon in underground spaces. The randomness of gas explosion accidents is intensified owing to naked fires, coal spontaneous combustion, electric sparks, metal surface impacts, or friction sparks that can cause the detonation of gases in coal mines.

The coal mine production system is a complex adaptive system, and the working environment is complex and dynamic [4]. There are exchanges of material, energy, and information resources among the subsystems in the system and with the external environment, and the circulation of material and energy flows is expressed in the form of information flow [5]. Therefore, it is necessary to conduct in-depth and systematic research on the causes of gas explosion accidents from the perspective of safety information transmission, and determine the fundamental causes of gas explosion accidents. In recent years, many studies have focused on safety information in the fields of network and food quality. This is still in its infancy in the domain of work safety. Scholars have studied the concept and connotation, flow process, and loss of safety information, but a complete system has not yet been developed [6,7,8,9].

Underwood and Waterson (2013) concluded that factors, such as information loss, have a major effect on system safety, and a high proportion of accidents are caused by information loss [10]. Zhao and Zhou (2012) stated that a lack of safety information would directly or indirectly lead to the occurrence of accidents and affect the development of accident scenarios and proposed an accident-causation theory and mathematical model of safety information loss [11]. Li et al. (2017) built a general model of safety information flow based on the information flow process and proposed a multilevel safety information asymmetry accident-causation model [12]. Liaw (2019) stated that safety information is the basic element of process safety management [13]. Feng (2019) built a safety information cognition (SIC) information flow model based on safety information cognition, constructed an SIC accident-causation model, and clarified the accident path [14]. Lei (2019) proposed an optimisation model of multilevel SIC that can optimise the transmission routes of safety information and reduce safety information distortion [15]. Guang (2020) constructed a PAR (Perceive-Analyze-Reply) accident-causation model based on safety information thinking, which can realise full-cycle and full-level accident prevention and analysis [16].

Therefore, based on the three elements of the source–channel–information subject and the six stages of information generation, conversion, perception, cognition, decision-making, and execution of safety information transmission, an accident-causation model was constructed in this study. Subsequently, the Decision-Making Trial and Evaluation Laboratory (DEMATEL) combined with the interpretive structural modelling (ISM) method was used to study the relationship between complex system factors. The causal factors of gas explosion accidents were analysed qualitatively and quantitatively, and the logical relationship between the causal factors was discussed to provide a theoretical basis for the effective prevention and reduction of gas explosion accidents in coal mines.

## 2. Safety Information Transmission

### 2.1. Safety Information

Li and Chen (2007) stated that safety information is a specific form that reflects the differences between safety features and their changes, including warning information, production status information, and safety instructions [17]. Zhao and Zhou (2012) proposed that safety information is all relevant information that can be known to humans, machines, and the environment in human production activities and can affect the occurrence and development of accidents. Sun and Hu (2014) stated that safety information is a collection of relevant data on production and life [18]. Wang and Wu (2017) defined safety information as the self-display of a future safety status [19]. Luo and Wu (2018) pointed out that safety information is a type of resource that can reflect the emergence, development, and change of all safety activities or entities in the field of safety. From the perspective of system safety, safety information refers to the safety status and changes in the system [20].

At present, the definition of safety information is not unified, but the essential attributes are consistent. It is mainly intended to protect the physical and mental safety of people, eliminate potential accidents, reduce accident losses, and ensure work safety. The relevant dataset reflecting the system safety state, such as safety warning signs, alarm information, and operation guidelines, are all safety information. The classification of safety information differs according to different classification standards. From the state of safety information, information can be divided into static and dynamic safety information. The safety information involved in underground production systems in coal mines includes safety signs, monitoring data, safety reports, rules, regulations, and daily records. Safety information is a specific representation of system safety. The identification of system hazards, occurrence of emergencies, development trends of the mastery, and intervention are all realised through the acquisition, analysis, and utilisation of safety information.

### 2.2. Safety Information Loss

Research on safety-information loss has mainly focused on safety-information asymmetry, missing safety information, and uncertainty in safety-information perception [21]. The essence of underground production in coal mines is the transmission and response to information. The process of information exchange and flow between organisation members, technical equipment, the environment, management, and other constituent elements in the production system is intricate. The information transmission process involves information sources, channels, and subjects. Among these, information sources provide safety information, which can be regarding either humans or mechanical equipment. The original information generated by the source must be transformed into data, sound waves, and light waves, and it is carried by the information carrier through the channel and perceived by the subject as perceptive information. After perceiving the information, the safety information subject needs to detect and convert the perceptive information by other complex physiological and psychological processes to form cognitive information, and then generate corresponding behavioural activities. Human behavioural activities are an external manifestation of cognitive information.

Safety information loss primarily occurs during information transmission. The entire process from information generation to information reception and use causes information loss, which includes information error, missing information, dislocation, and delay. Safety information transmission mainly involves transmission between the information source and subject as well as transmission between the information subject and subject. Among these, there is a large amount of signal noise in the transmission process between the information source and information subject. At the same time, owing to the chaotic crossover of channels, conflict, and collision, confusion can easily occur when people receive information, which affects the accuracy and timeliness of information transmission. In addition to poor channel capabilities and other reasons, the information transmission rate is low, resulting in the distortion of the information transmission. In the process of information transmission between information subjects, owing to differences in the cognitive ability, understanding, and environment of people, they may selectively receive information, which inevitably results in information loss. The degree of information loss directly affects the possibility of accidents. Information transmission and subject responses are key links to ensure the spread of original information. Information distortion is more serious when the information transmission link is longer. For higher comprehension quality of the information-receiving subject, the utilisation rate of the information is higher.

## 3. Coal Mine Gas Explosion Accidents

### 3.1. Mechanism of Gas Explosion Accidents

A gas explosion can occur if the three following conditions are met simultaneously: (1) The gas concentration must be within the explosive limit (generally 5–16% CH4), (2) the oxygen content in the air must be higher than 12%, and (3) a high-temperature ignition source (generally >650 °C) must be present. In underground places, except for blind lanes, the oxygen concentration reaches 20%. Therefore, the prevention of gas explosion accidents must be controlled based on two aspects: the gas concentration and ignition source. Claus postulated that information is the product of data processed by human consciousness. Gas concentration data is used as safety information to prevent gas explosions. Once the limit is exceeded and not detected in time, safety information is lost. Gas concentration monitoring involves tile inspectors, workers, safety supervisors, etc. Gas concentration overrun is the direct cause of gas explosion accidents, and the causes of overrun mainly include ventilator failure, insufficient air volume, chaotic ventilation systems, and inadequate implementation of tile inspection systems. The fundamental reason for this is deviation in safety-information acquisition, monitoring, and transmission. Missing information and information asymmetry, caused by deviations in the transmission of system safety information (information loss, incorrect information, and abnormal information flow), are the main causes of gas explosion accidents.

### 3.2. Causes of Gas Explosion Accidents

Through a large number of accident cases and theoretical analyses, it can be correctly stated that safety information is the specific manifestation of the mechanism and influencing factors of gas explosion accidents. The essential phenomenon of the accident evolution path and logical relationship is the transmission of safety information. This includes gas concentration monitoring, oxygen content, and ignition sources, which are categorized as dynamic, explicit, and objective safety information. These is also key safety information to prevent gas explosion accidents. Critical information loss is the root cause of such accidents. Based on the accident mechanism, the reasons and factors that cause critical information loss are analysed at the macro and micro levels. The macro level is mainly at the organisational level, including the social environment and coal mine management. The micro-level is mainly at the individual level, including the behaviour of miners and the state of objects.

Considering the macro-level of coal mine production systems, gas explosion accidents are reduced with the progress of society and technological innovation, strengthening of government regulation, social supervision, and enterprise management. According to on-site research and interviews, there are differences between the implementation documents issued by the government and the actual conditions of the coal mines. The phenomenon of mismatch of technical measures, unequal information, and simple rote writing is common. There are many problems in the information transmission process, and information quality is the basic condition to ensure the effectiveness of information transmission. If the information is misinterpreted or the original information is incorrect, the accident risk can increase from the root.

Considering the micro-level of coal mine production systems, human factors cause fire source and gas accumulation that leads to coal mine gas explosions. Unsafe behaviour by miners is the main cause of gas accumulation and the presence of fire sources. From the perspective of informatics, gas concentration is the source of information, which is obtained through equipment detection. The information loss in this stage mainly involves two aspects: the frequency of the concentration detection and the accuracy of the detection equipment. However, regarding concentration prediction, once gas accumulation occurs, if the concentration is not detected in time to make a judgment, there can be potential explosion hazards.

## 4. Accident-Causation Model of Coal Mine Gas Explosion

### 4.1. Analysis of Influencing Factors of Safety Information Loss

In the entire process of safety-information generation, conversion, perception, cognition, decision-making, and execution transmission, the information source, channel, and subject are the key constituent elements of information transmission. In the process of information flow, information carrier defects or limited function, information conversion, external interference, insufficient information stimulation intensity, and time duration cause information loss, distortion, delay, and dislocation. Information loss ultimately affects the behavioural response of workers. Therefore, information loss in the process of information transmission can directly or indirectly lead to the occurrence of gas explosion accidents and affect their further expansion. Avoiding crucial safety information loss is key to preventing the occurrence and expansion of accidents.

There are many types of safety information in underground coal mine production systems, and dynamic information is complex and changeable. Safety information is the key connection point to ensure the orderly and stable operation of the system. In the entire process from information generation to decision transmission, the factors and causes of safety information loss can be analysed from three constituent elements of information transmission (information source, information channel, and information subject) and six main processes (generation, conversion, perception, cognition, decision-making, and execution). The constituent elements of information transmission are the basic elements that ensure the authenticity of information. Once a certain element fails or has defects, it leads to information loss, thus affecting the normal operation of the system. The six main processes of information transmission are key links that affect the degree of information loss. A critical point in the degree of information loss that leads to an accident is called crucial safety information. Crucial safety information refers to the minimum amount of information required to ensure safe operation of the system. Once crucial safety information is transmitted with error, missing data, delay, dislocation, and other losses, it can affect the safety prediction, decision-making, and other follow-up links of the information subject. Decision-making errors can lead to unsafe behaviour by workers and unsafe state of equipment, which can cause accidents.

By analysing the information sources, channels, and subjects involved in the entire process of safety information transmission, it is evident that the defects in safety information sources develop into faults under the influence of the outside environment, resulting in inaccurate and false information release. Channel failure due to external influences results in poor channel and channel interruptions, and safety information cannot be transmitted normally. Information subjects cannot receive accurate safety information, and cannot make correct predictions, decisions, and execution behaviour, resulting in increased possibility of accidents.

From the perspective of the entire process of information transmission, changes in the safety information quality, safety information dissemination environment, and safety information subjects all affect the effectiveness and reliability of information, which are the main factors causing information loss, as shown in Figure 1.

Safety information quality is the basic condition for the effective transmission of information, including the reliability of information sources, clarity of purpose, normative information, and timeliness. If the information itself contains errors or is invalid, it can directly affect the decision-making behaviour of people, which can lead to accidents. In the process of effective information transmission, owing to the impact of untimely dissemination, signal-to-noise destruction, channel diversity, and other environmental influences, there is deviation or loss in the information transmission process. Therefore, the information-dissemination environment significantly affects the transformation and transmission of safety information. The carrier, manifestation, and time of information transmission mainly include dissemination timeliness, signal-noise destructiveness, diversity expression, system complexity, and channel diversity. The perception, cognition, and decision-making process of information receivers for safety information are affected by the level of personal safety awareness, emergency response ability, safety knowledge and skills, physical and mental conditions, and work experience. This is closely related to the overall quality of an individual.

### 4.2. Construction of Accident-Causation Model

Studies have shown that unsafe behaviour is the root cause of accidents, and the loss of safety information is the main factor that causes unsafe behaviour, which plays a decisive role in the occurrence of accidents [22,23]. Therefore, based on the three elements of source, channel, and information subject, in this study, we analyse the generation, conversion, perception, cognition, decision-making, and execution of information transmission and propose a causal model of coal mine gas explosion accidents based on safety information transmission, which provides new ideas and methods for the prevention of gas explosion accidents. This causal model postulates that information loss in the process of information transmission plays a decisive role in gas explosion accidents, which is not only the potential inducement of the accident, but also the main reason for the expansion of the gas explosion accident, as shown in Figure 2.

The occurrence of gas explosion accidents is accompanied by the whole process of information transmission. Safety information loss, such as inadequate information transmission, neglect of safety (danger) information, deviation of information perception, and incorrect or untimely information responses, are the key factors that lead to gas explosion accidents. This includes the actions of neglecting safety information, such as the failure to take response measures in time when the gas exceeds the limit, failure to obtain timely gas concentration information, the chaotic nature of the ventilation system, and the wrong operation process. The essence of accident prevention and control is to control, regulate, and manage the material, energy, and information flow in the system through the identification, guidance, monitoring, warning, and regulation of safety information (positive effect). Safety information that wrongly reflects the state of material flow and energy flow in the system can lead to accidents or expand the impact of accidents (negative effect). Therefore, it is necessary to give maximum importance to and correctly use the guidance and control effect of safety information flow on material flow and energy flow to ensure the efficient transmission and timely and correct response to safety information.

## 5. Quantitative Analysis Based on DEMATEL-ISM

The DEMATEL method is a methodology for solving complex problems proposed by American scholars Gabus and Fontela. It calculates the degree of influence of each factor on other factors by determining the size of the direct influence relationship between the factors in the system. It obtains the centrality and causation degree of each factor to determine the cause and result factors [24]. The ISM method decomposes a complex system into several subsystems and, ultimately, forms a multilevel hierarchical interpretative structure model. The integration of DEMATEL and ISM methods can effectively reduce the computational volume and computational complexity of the Accessibility matrix, and facilitate the scientific analysis of the influencing factor system. From a quantitative point of view, the causes of gas explosion accidents were analyzed based on DEMATEL-ISM. The logical relationship, causal relationship and attribute characteristics between various risk factors that lead to the loss of safety information are deeply discussed, so as to provide theoretical support for the prevention and control of gas explosion accidents.

### 5.1. Correlation Characteristic Analysis

#### 5.1.1. Identification of Causal Factors in Gas Explosion Accidents

The construction of the accident-causation model of gas explosion accidents, is based on the influencing factors of safety information loss. These include hidden danger information leading to gas explosion accidents, and the impact of the external environment, including gas concentration overruns, ignition sources, government regulations, and social supervision. The causal factors of gas explosions based on safety information transmission are listed in Table 1.

#### 5.1.2. Establishment of Direct Influence Matrix *K*

Eight experts in the field of coal mine safety and emergency management were invited to participate and assess, according to the scale 0–3 (0: no impact; 1: weak impact; 2: medium impact; 3: strong impact). The scoring method was used to quantitatively evaluate the direct influence relationship and degree of influence among various factors of gas explosion, and the direct influence matrix is shown in Table 2.
(1)$$K={\left[k_{ij}\right]}_{18\times 18}$$
where $k_{ij}$ indicates the influence degree of the factor i on the factor j. As shown in Table 2, where $k_{34}=1$, it implies that the influence of C3 on C4 is weak.

#### 5.1.3. Establishment of Comprehensive Influence Matrix *T*

① Standardize the direct influence matrix *K* to form the standardized direct influence matrix *N*.
(2)N=K/s
(3)$$s=\max\left[\underset{1\leq i\leq n}{\max}\sum_{i=1}^{n}k_{\mathrm{ij}},\underset{1\leq i\leq n}{\max}\sum_{j=1}^{n}k_{\mathrm{ij}}\right]$$
where s indicates the maximum value of the sum of rows and the sum of columns.

② To further analyse the indirect influence relationship and degree of influence of the causal factors of gas explosions, a comprehensive influence matrix *T* is constructed based on Equation (4).
(4)$$T=N{\left(I-N\right)}^{-1}$$
where *I* indicates the unit matrix of the same order as *N*.

#### 5.1.4. Determination of Centrality and Causation Degree

The influence degree $r_{i}$ of the corresponding factors is obtained by the sum of the rows in the comprehensive influence matrix *T*, and the degree $c_{i}$ of the corresponding factors is obtained by the sum of the columns in the comprehensive influence matrix *T*; $r_{i}+c_{i}$ indicates the centrality degree, and $r_{i}-c_{i}$ indicates the causation degree, as shown in Table 3. The centrality degree refers to the importance of factors in the system, and the causation degree refers to the degree to which factors affect other factors. If the degree of centrality is greater, there is a more significant role of highly impactful factors in gas explosion accidents. For a larger degree of causation, there is stronger correlation between factors. When the causation degree is greater than zero, it shows that factors have a greater impact on other factors, and are called cause factors. When the causation degree is less than 0, it shows that the factors are strongly affected by other factors, and are called result factors. The result factors are the comprehensive embodiment of the role of causal factors.

It is evident from Table 3 that the centrality degrees of C4 (information timeliness), C15 (gas concentration overrun), C17 (Government regulation), C5 (dissemination timeliness), C16 (ignition source appears), C10 (safety awareness level), C9 (channel diversity), and C2 (purpose readability) were relatively high, indicating that they had a significant influence on the occurrence of gas explosion accidents and occupied a core position among other factors. Significant attention must be paid to prevent gas explosion accidents. Safety information is a form of expression of the state of the system, and the safety and danger of the current state can be understood by mastering the interpretation of information. Therefore, the timeliness of information itself and the timeliness of information transmission play an important role in preventing gas explosion accidents, especially critical safety information, such as gas concentration overrun and fire source information.

As shown in Table 3, C8 (system complexity), C14 (work experience), C10 (safety awareness level), C18 (social supervision), C17 (Government regulation), C6 (signal noise severity), C9 (channel diversity), C7 (diversity expression), C13 (physical and mental), and C12 (safety knowledge and skill) were the causal factors, among which C8 had the greatest impact on other factors. This indicated that system complexity was the key to information transmission loss and gas explosion accidents. In addition, C2 (purpose readability), C5 (dissemination timeliness), C15 (gas concentration overrun), C16 (ignition source appears), C4 (information timeliness), C3 (information normative), C1 (source reliability) and C11 (emergency handling ability) were the result factors, among which C2, C5, C15, and C16 were strongly affected by other factors. The factors affecting C2, C5, C15, and C16 must be actively explored and controlled.

### 5.2. Hierarchical Characteristic Analysis

#### 5.2.1. Adjacency Matrix *L* and Accessibility Matrix *Z*

The adjacency matrix *L* was constructed according to the comprehensive influence matrix *T*, and the adjacency matrix *L* was added to the unit matrix *I* to form a new matrix *H* based on Equation (6). Then, according to the operation properties of Boole, MATLAB software was used to perform multiple Boolean operations on matrix *H* until the formula $Z={\left(L+I\right)}^{n+1}={\left(L+I\right)}^{n}\neq {\left(L+I\right)}^{n-1}\neq L+I$ was satisfied, and finally the accessibility matrix *Z* was obtained, as shown in Table 4.
(5)$$l_{ij}=\left\{\begin{array}{l} 0,l_{ij}<\lambda \\ 1,l_{ij}\geq \lambda \end{array}\right.$$
(6)H=L+I
where $l_{ij}$ indicates the element of the adjacency matrix *L*. λ indicates the boundary threshold, which is the sum of the average value and standard deviation of all influence values in the comprehensive influence matrix *T*.

#### 5.2.2. Multilevel Hierarchical ISM

According to the hierarchy principle, the factor satisfying $C\left(S_{i}\right)=P\left(S_{i}\right)$ was extracted as the first-level factor. The rows and columns corresponding to this factor were then deleted from the accessibility matrix *Z*, and this process was repeated to divide the accessibility matrix *Z* into structural levels [25]. By constructing a multilayer hierarchical structure model, the hierarchical structural relationship between the causal factors of gas explosion accidents was determined, and the root factors, surface factors, and transition factors of gas explosion accidents were clarified, as shown in Figure 2.
(7)$$P\left(S_{i}\right)=\left\{S_{i}|S_{i}\in Z,z_{ij}=1\right\},\left(i=1,2,\ldots ,n\right)$$
(8)$$Q\left(S_{j}\right)=\left\{S_{j}|S_{j}\in Z,z_{ji}=1\right\},\left(j=1,2,\ldots ,n\right)$$
(9)$$C\left(S_{i}\right)=P\left(S_{i}\right)$$
where $P\left(S_{i}\right)$ indicates the reachable set of the accessibility matrix *Z*, $Q\left(S_{j}\right)$ indicates the antecedent set of the accessibility matrix *Z*, $C\left(S_{i}\right)$ indicates the common set, S indicate the factors of the accessibility matrix *Z*.

Through hierarchical division, the hierarchical structure and correlation characteristics of the factors influencing gas explosions were clarified. Regarding the causal factors of gas explosion accidents, more attention should be paid to higher hierarchical structure levels. As shown in Figure 3, C8 (system complexity) was the root factor of gas explosion accidents and had a profound impact on other factors. C14 (work experience) and C17 (Government regulation) were in the sixth layer, which were affected by the seventh layer factors, and were also the factors affecting other layers. These affected the occurrence of gas explosion accidents. In addition, C2 (purpose readability) and C16 (ignition source appearance) were surface factors. The intermediate level was the transition factor, and the root factors acted on the surface factors by affecting them.

## 6. Discussion

(1)The information subject is not only the recipient of safety information, but also the transmitter of information. Work experience, safety awareness level, and the physical and mental state of information subjects have a significant impact on information transmission, which directly affects the accuracy and effectiveness of information transmission and use. Therefore, coal mining enterprises need to improve the safety awareness level, information cognition, and processing ability of their employees through daily safety training and assessment. In addition, signal noise, dissemination channels, and information expression forms in the information transmission process have a certain impact on information loss. It is necessary to consider the ways and manifestations of information transmission from the perspective of information types, the needs of the information subjects, and understanding. This can help avoid the homogenization of information transmission and further improve the effectiveness of information transmission.(2)Owing to the complexity of underground production systems, there are many types and structures of safety information in the coal mine production process. There are problems in the quality assurance of the source, the transmission environment is complex and changeable, and the comprehensive quality of the information recipients is uneven, which leads to problems in the process of information transmission. Different degrees of loss result in gas accumulation or the emergence of fire sources, which leads to gas explosion accidents. Therefore, in view of the complexity of the system, it is necessary to comprehensively analyse the information types, characteristics, attributes, and target states. Additionally, the safety information should be divided according to the urgency, manifestation, and response needs. The information transmission efficiency and information utilisation rate must be improved, and the role of safety information must assume full importance.

## 7. Conclusions

In this study, we analysed the entire process of safety information transmission and postulate that the loss of critical safety information is the main reason for accidents. Therefore, improving the quality of the source, improving the information transmission environment, and strengthening the comprehensive quality of the information recipients can ensure the accuracy and timeliness of safety information transmission and avoid the loss or misuse of critical safety information. In this study, we provide a new idea and method for the prevention of gas explosion accidents.

(1)Loss of safety information is a key factor in gas explosion accidents. Based on the entire process of safety information transmission, the factors affecting safety information loss were extracted, including safety information quality, safety information environment, safety information subjects, three first-level indicators, and fourteen second-level indicators.(2)The three elements (source, channel, and information subject) and six stages (generation, conversion, perception, cognition, decision-making, and execution) of safety information transmission were analyzed. This was combined with the case of gas explosion accidents to develop an accident-causation model of a gas explosion accident based on safety information transmission, which has a certain guiding significance for the prevention of gas explosion accidents.(3)Eighteen causal factors were extracted according to the accident-causation model of a gas explosion, based on safety information transmission. The DEMATEL-ISM quantitative analysis method was used to determine ten cause factors and eight result factors. Among them, C8, C14, and C17 were the root factors affecting the occurrence of gas explosion accidents, and C2 and C16 were the surface factors.

The loss of critical safety information is the fundamental factor leading to accidents. This paper analysed the causes of gas explosion accidents based on the whole process of safety information transmission, but lacks a quantitative assessment of safety information loss. In the future, we will continue to classify and extract the safety information of various types of accidents in coal mines, and quantify the loss in the transmission process, and conduct further in-depth research on coal mine accidents from the perspective of the amount of safety information loss.

## Figures and Tables

**Figure 1 ijerph-19-10126-f001:**
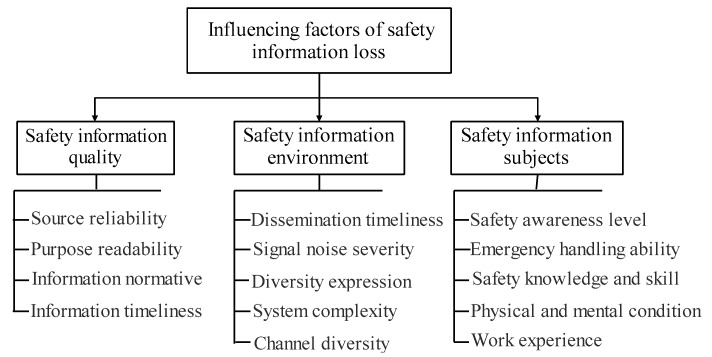
Influencing factors of safety information loss.

**Figure 2 ijerph-19-10126-f002:**
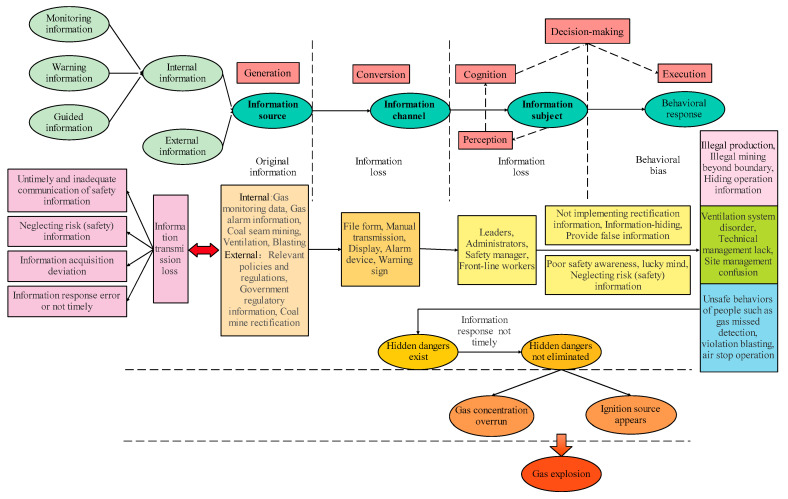
Accident-causation model of coal mine gas explosion based on safety information transmission.

**Figure 3 ijerph-19-10126-f003:**
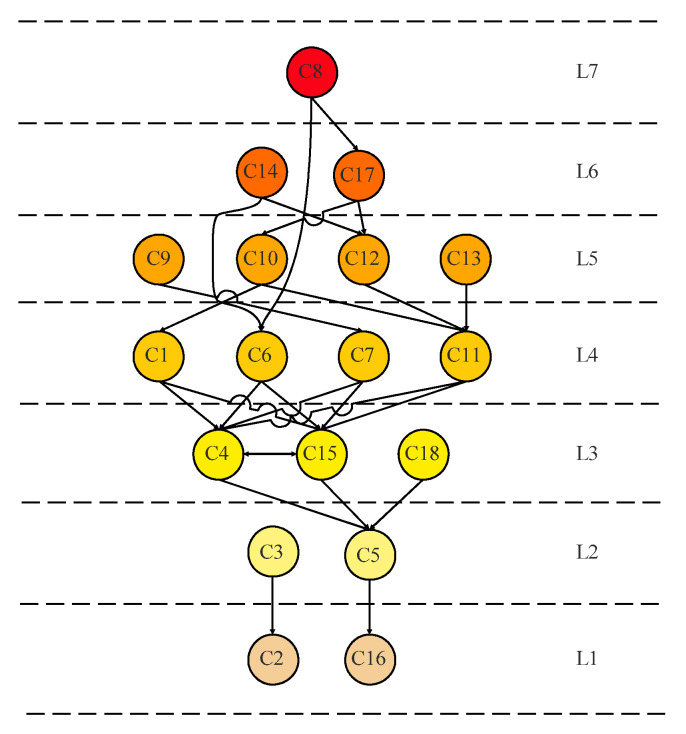
Multilevel hierarchical interpretative structure model of gas explosion accidents.

**Table 1 ijerph-19-10126-t001:** Causal factors of gas explosion accidents.

Causal Factors		Causal Factors		Causal Factors	
Safety information quality	Source reliability C1	Safety information environment	Dissemination timeliness C5	Safety information subjects	Safety awareness level C10
	Purpose readability C2		Signal noise severity C6		Emergency handling ability C11
	Information normative C3		Diversity expression C7		Safety knowledge and skill C12
	Information timeliness C4		System complexity C8		Physical and mental condition C13
Hidden dangers	Gas concentration overrun C15		Channel diversity C9		Work experience C14
	Ignition source appears C16	External environment	Government regulation C17	Social supervision C18	

**Table 2 ijerph-19-10126-t002:** Direct influence matrix.

Factors	C1	C2	C3	C4	C5	C6	C7	C8	C9	C10	C11	C12	C13	C14	C15	C16	C17	C18
C1	0	1	2	0	0	0	0	1	1	0	0	0	0	0	3	1	0	0
C2	0	0	0	0	0	0	0	0	0	0	1	1	0	0	1	1	0	0
C3	1	3	0	1	1	0	1	0	0	0	1	1	0	0	1	0	0	0
C4	0	0	0	0	3	1	0	0	2	0	0	0	0	0	3	3	2	1
C5	0	0	0	2	0	0	1	0	1	0	0	0	0	0	2	3	0	0
C6	0	3	3	3	3	0	0	0	1	0	1	0	2	0	3	3	2	0
C7	0	2	1	2	2	1	0	1	2	0	1	0	0	0	1	1	2	1
C8	0	3	3	3	3	3	0	0	1	1	2	2	2	0	3	3	3	1
C9	0	3	0	3	3	2	3	0	0	0	1	0	0	0	3	3	2	1
C10	2	3	2	3	3	0	1	0	1	0	2	2	1	0	3	3	2	0
C11	0	0	0	1	3	0	1	0	0	1	0	2	0	0	3	3	0	0
C12	2	3	2	0	1	0	0	0	0	2	3	0	1	0	3	3	1	0
C13	2	2	2	2	3	2	0	0	0	2	3	1	0	0	2	2	0	0
C14	1	2	2	2	2	3	0	0	1	2	3	3	1	0	2	2	2	1
C15	2	2	1	3	2	0	0	0	0	0	0	0	2	0	0	0	0	0
C16	2	2	2	2	2	0	0	0	0	0	0	0	2	0	0	0	0	0
C17	3	3	3	3	2	0	1	0	2	3	3	3	1	0	1	1	0	1
C18	1	2	1	1	2	0	2	0	2	2	1	1	2	0	0	0	2	0

**Table 3 ijerph-19-10126-t003:** Comprehensive influence matrix.

Factors	C1	C2	C3	C4	C5	C6	C7	C8	C9	C10	C11	C12	C13	C14	C15	C16	C17	C18	$r_{i}+c_{i}$	$r_{i}-c_{i}$
C1	0.01	0.06	0.07	0.03	0.02	0.01	0.01	0.03	0.03	0.00	0.01	0.01	0.01	0.00	0.11	0.05	0.01	0.00	1.42	−0.48
C2	0.01	0.01	0.01	0.01	0.01	0.00	0.00	0.00	0.00	0.00	0.03	0.03	0.01	0.00	0.04	0.04	0.00	0.00	2.02	−1.60
C3	0.04	0.11	0.01	0.05	0.05	0.00	0.03	0.00	0.01	0.00	0.04	0.04	0.01	0.00	0.05	0.02	0.01	0.00	1.70	−0.76
C4	0.03	0.04	0.03	0.05	0.13	0.04	0.02	0.00	0.08	0.01	0.02	0.01	0.02	0.00	0.13	0.13	0.07	0.04	2.58	−0.90
C5	0.01	0.02	0.01	0.08	0.03	0.01	0.04	0.00	0.04	0.00	0.01	0.00	0.01	0.00	0.08	0.11	0.01	0.01	2.42	−1.46
C6	0.04	0.14	0.12	0.15	0.15	0.01	0.02	0.00	0.05	0.02	0.06	0.02	0.08	0.00	0.15	0.14	0.08	0.01	1.76	0.70
C7	0.02	0.10	0.06	0.11	0.11	0.04	0.02	0.03	0.08	0.01	0.05	0.02	0.02	0.00	0.08	0.08	0.08	0.04	1.53	0.41
C8	0.05	0.18	0.15	0.18	0.19	0.11	0.03	0.00	0.06	0.06	0.11	0.10	0.10	0.00	0.18	0.18	0.12	0.04	1.93	1.71
C9	0.03	0.14	0.04	0.15	0.16	0.07	0.11	0.00	0.03	0.01	0.06	0.02	0.03	0.00	0.15	0.15	0.09	0.04	2.03	0.53
C10	0.10	0.15	0.10	0.15	0.16	0.01	0.05	0.00	0.06	0.02	0.09	0.08	0.06	0.00	0.16	0.16	0.08	0.01	2.06	0.84
C11	0.02	0.03	0.02	0.07	0.12	0.01	0.04	0.00	0.01	0.04	0.01	0.07	0.02	0.00	0.12	0.12	0.01	0.00	1.80	−0.34
C12	0.09	0.14	0.09	0.05	0.08	0.01	0.01	0.00	0.01	0.07	0.11	0.02	0.05	0.00	0.14	0.13	0.04	0.00	1.85	0.27
C13	0.09	0.11	0.09	0.11	0.15	0.07	0.02	0.00	0.02	0.07	0.11	0.05	0.02	0.00	0.12	0.12	0.02	0.01	1.95	0.41
C14	0.08	0.14	0.11	0.14	0.15	0.10	0.02	0.00	0.06	0.09	0.13	0.12	0.07	0.00	0.15	0.15	0.09	0.04	1.64	1.64
C15	0.07	0.08	0.04	0.11	0.09	0.01	0.01	0.00	0.01	0.01	0.01	0.01	0.07	0.00	0.03	0.03	0.01	0.00	2.49	−1.31
C16	0.07	0.08	0.07	0.08	0.08	0.01	0.01	0.00	0.01	0.01	0.01	0.01	0.07	0.00	0.03	0.03	0.01	0.00	2.39	−1.23
C17	0.13	0.16	0.13	0.15	0.14	0.02	0.06	0.01	0.09	0.11	0.13	0.12	0.06	0.00	0.12	0.11	0.03	0.04	2.45	0.77
C18	0.06	0.11	0.06	0.08	0.12	0.01	0.08	0.00	0.08	0.08	0.07	0.05	0.08	0.00	0.06	0.06	0.08	0.01	1.41	0.81

**Table 4 ijerph-19-10126-t004:** Accessibility matrix.

Factors	C1	C2	C3	C4	C5	C6	C7	C8	C9	C10	C11	C12	C13	C14	C15	C16	C17	C18
C1	1	0	0	1	1	0	0	0	0	0	0	0	0	0	1	1	0	0
C2	0	1	0	0	0	0	0	0	0	0	0	0	0	0	0	0	0	0
C3	0	1	1	0	0	0	0	0	0	0	0	0	0	0	0	0	0	0
C4	0	0	0	1	1	0	0	0	0	0	0	0	0	0	1	1	0	0
C5	0	0	0	0	1	0	0	0	0	0	0	0	0	0	0	1	0	0
C6	0	1	1	1	1	1	0	0	0	0	0	0	0	0	1	1	0	0
C7	0	1	0	1	1	0	1	0	0	0	0	0	0	0	1	1	0	0
C8	1	1	1	1	1	1	0	1	0	1	1	1	1	0	1	1	1	0
C9	0	1	0	1	1	0	1	0	1	0	0	0	0	0	1	1	0	0
C10	1	1	0	1	1	0	0	0	0	1	1	0	0	0	1	1	0	0
C11	0	0	0	1	1	0	0	0	0	0	1	0	0	0	1	1	0	0
C12	0	1	0	1	1	0	0	0	0	0	1	1	0	0	1	1	0	0
C13	0	1	0	1	1	0	0	0	0	0	1	0	1	0	1	1	0	0
C14	0	1	1	1	1	1	0	0	0	0	1	1	0	1	1	1	0	0
C15	0	0	0	1	1	0	0	0	0	0	0	0	0	0	1	1	0	0
C16	0	0	0	0	0	0	0	0	0	0	0	0	0	0	0	1	0	0
C17	1	1	1	1	1	0	0	0	0	1	1	1	0	0	1	1	1	0
C18	0	1	0	0	1	0	0	0	0	0	0	0	0	0	0	1	0	1

## Data Availability

We declare that all data, models, and code generated or used during the study appear in the submitted article.

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
