# Peer review of "Research on the Causes of Gas Explosion Accidents Based on Safety Information Transmission"

_ijerph, 2022, doi:10.3390/ijerph191610126_

Round 1

Reviewer 1 Report

The presented paper shows a quantitative analysis of gas explosion accidents using information transmission framework. The subject of this study is interesting and has certain theoretical and practical application value. The recommendation is minor revision with the following comments.

(1)    The text explaining equations (7), (8) and (9) cannot be understood. There are unclear variables with no explanation whatsoever. If the reader shall be able to follow the computation, more clear explanation needs to be added. Or at least include reference to other publication, where the reader can understand the computation executed in the paper.

(2)    The limitations of the study are not realized. They should be listed in the Conclusions and Future Work section and some directions for future work should be mentioned.

Author Response

Q1: The text explaining equations (7), (8) and (9) cannot be understood. There are unclear variables with no explanation whatsoever. If the reader shall be able to follow the computation, more clear explanation needs to be added. Or at least include reference to other publication, where the reader can understand the computation executed in the paper.

A1: Thank you very much for your comment. We have added the explanation of some unclear variables of equations (7), (8) and (9). In addition, in order to better understand the computation executed in the paper, we have added the reference to other publication.

[26] Li, G.L.; Yan, Y.Z.; Liu, W.Q.; et al. Research on formation factors of miners’ unsafe emotions based on DMATEL-ISM. China Safety Science Journal. 2021, 31(7),30-37.

Q2: The limitations of the study are not realized. They should be listed in the Conclusions and Future Work section and some directions for future work should be mentioned.

A2: Thank you very much for your comment, we have added the limitations of the study and some directions for future work in line 448-454.

      The loss of critical safety information is the fundamental factor leading to accidents. This paper analyzes the causes of gas explosion accidents based on the whole process of safety information transmission, but lacks a quantitative assessment of safety information loss. In the future, we will continue to classify and extract the safety information of various types of accidents in coal mines, and quantify the loss in the transmission process, and conduct further in-depth research on coal mine accidents from the perspective of the amount of safety information loss.

Reviewer 2 Report

In my opinion, the manuscript is generally well constructed, but is stress misinformation as the main cause of accidents, when accidents are multicausal phenomena. I would stress in the text that misinformation is a factor which strongly influences other causes of accidents in coal mines. Moreover, points 1 to 3 are in my opinion too prolix and discursive, I suggest simplifying them.

A reference is needed for estatement of lines 251 -3.

Author Response

Q1: In my opinion, the manuscript is generally well constructed, but is stress misinformation as the main cause of accidents, when accidents are multicausal phenomena. I would stress in the text that misinformation is a factor which strongly influences other causes of accidents in coal mines. Moreover, points 1 to 3 are in my opinion too prolix and discursive, I suggest simplifying them.

A1: Thank you very much for your comment, this paper explains the impact of misinformation on accidents in part 4.1. Safety information quality is the basic condition for the effective transmission of information, including the reliability of information sources, clarity of purpose, normative information, and timeliness. If the information itself contains errors or is invalid, it can directly affect the decision-making behaviour of people, which can lead to accidents.

In addition, I'm really sorry that points 1 to 3 you mentioned are too prolix and discursive, which part is exactly, I'm not very clear here.

Q2: A reference is needed for estatement of lines 251-3.

A2: Thank you very much for your comment. We have added the reference for estatement of lines 251-3.

[23] Huang, L.; Wu, C.; Wang, B. Individual Behavioral Safety Mechanism and Its Influence Factors Based on Information Cognition. Journal of Intelligence. 2018, 37(8), 121-127.

[24] Cheng, L.H.; Jiang, B.L.; Guo, H.M. Modeling the causes of accidental gas explosions from the perspective of safety information loss. Process Safety Progress. 2022.

Reviewer 3 Report

The review document is attached.

Author Response

Q1: DEMATEL methodology for solving complex problems has been reviewed in 2018 in the paper: “Sheng-Li Si, Xiao-Yue You, Hu-Chen Liu, Ping Zhang, "DEMATEL Technique: A Systematic Review of the State-of-the-Art Literature on Methodologies and Applications", Mathematical Problems in Engineering, vol. 2018, Article ID 3696457, 33 pages,2018.https://doi.org/10.1155/2018/3696457”. It is advisable to include this paper among references, for readers who do not know this methodology.

A1: Thank you very much for your comment, we have added this reference in the paper.

[25] Si, S.L.; You, X.Y.; Liu, H.C.; et al. DEMATEL Technique: A Systematic Review of the State-of-the-Art Literature on Methodologies and Applications. Mathematical Problems in Engineering. 2018, 2018,1-33.

Q2: What stands for the acronym ISM?

A2: Thank you very much for your comment. ISM is the acronym for interpretive structural modelling, which is explained in line 74 of this article.

Typos: Throughout the paper: “timeliness” should be “timelines”.

       After full consideration, according to the specific meaning expressed in this article, we believe that timeliness should be more appropriate to represent the timeliness of information transmission.
